

# Role of the water balance constraint in the long short-term memory network: large-sample tests of rainfall-runoff prediction

Qiang Li[1], Tongtiegang Zhao[1]

[1]Southern Marine Science and Engineering Guangdong Laboratory (Zhuhai), Key Laboratory for Water Security in the
Guangdong-Hongkong-Macao Greater Bay Area, School of Civil Engineering, Sun Yat-Sen University, Guangzhou, China,

*Correspondence to*: Tongtiegang Zhao (zhaottg@mail.sysu.edu.cn)

**Abstract.** While deep learning (DL) models are effective in rainfall-runoff modelling, their dependence on data and lack of
physical mechanisms can limit their use in hydrology. As there is yet no consensus on the consideration of the fundamental
water balance for DL models, this paper presents an in-depth investigation of the effects of water balance constraint on the
long-short term memory (LSTM) network. Specifically, based on the Catchment Attributes and Meteorology for Large-sample
Studies (CAMELS) dataset, the LSTM and its architecturally mass-conserving variant (MC-LSTM) are trained basin-wise to
provide rainfall-runoff prediction and then the robustness of the LSTM and MC-LSTM against data sparsity, random
parameters initialization and contrasting climate conditions are assessed across the contiguous United States. Through large-
sample tests, the results show that the water balance constraint evidently improves the robustness of the basin-wise trained
LSTM. On the one hand, as the amount of training data increases from 1 year to 15 years, the incorporation of the water
balance constraint into the LSTM network decreases the sensitivity from 95.0% to 32.7%. On the other hand, the water balance
constraint contributes to the stability of the LSTM for 450 (85%) basins when there are 3 years' training data. In the meantime,
the water balance constraint improves the transferability of the LSTM from the driest years to the wettest years for 318 (67%)
basins. Overall, the in-depth investigations of this paper facilitate insights into the use of DL models for rainfall-runoff
modelling.





**Short summary.** The lack of physical mechanism is a critical issue for the use of popular deep learning models. This paper presents an in-depth investigation of the fundamental mass balance constraint for deep learning-based rainfall-runoff prediction. The robustness against data sparsity, random parameters initialization and contrasting climate conditions are detailed. The results highlight that the water balance constraint evidently improves the robustness in particular when there is limited training data.



## 1 Introduction

Deep learning (DL) has been increasingly used for rainfall-runoff modelling (Kratzert et al., 2018; Lees et al., 2021;
Nearing et al., 2021; Shen, 2018; Tsai et al., 2021). Without explicit descriptions of the underlying physical processes and
related assumptions, DL models are set up to directly capture response patterns hidden in large datasets (Feng et al., 2020;
LeCun et al., 2015). DL models have been shown to exhibit superiority in effectively simulating complex nonlinear systems
across different fields owing to the rapid growth of available data and advances in computational capability (LeCun et al.,
2015; Reichstein et al., 2019; Wang et al., 2023). Effective in dealing with the complexity and nonlinearity of rainfall-runoff
processes, DL models have become popular in hydrological applications (Frame et al., 2022; Gauch et al., 2021a; Nearing et
al., 2021; Kratzert et al., 2018). There are extensive uses of the long short-term memory (LSTM) network (Kratzert et al.,
2018), the recurrent neural network (Nagesh Kumar et al., 2004), the gate recurrent unit (Zhang et al., 2021), the sequence-to-
sequence model (Xiang et al., 2020) and the encoder-decoder model (Kao et al., 2020).

The LSTM network is one of the most important DL models (Feng et al., 2021; Jiang et al., 2022; Kao et al., 2020; Lees
et al., 2022; Razavi, 2021). Due to the recurrent structure and unique gating mechanism (Hochreiter and Schmidhuber, 1997),
the LSTM network can account for not only nonlinear relationships but also temporal dependencies among variables (Jiang et
al., 2022; Read et al., 2019). These inherent capabilities make the LSTM network well suited for modelling hydrologic
dynamics, especially multi-scale memory effects such as the persistence and release of water from soil moisture and snowpack
(Pokharel et al., 2023; Wi and Steinschneider, 2022). To date, substantial efforts have been made to exploit the predictive
capability of the LSTM network (Jiang et al., 2022). Compared to process-based hydrologic models, the LSTM network has
been shown to be similarly effective or even better in rainfall-runoff prediction (Gauch et al., 2021a; Lees et al., 2021; Kratzert
et al., 2018). There were thorough tests of Predictions in Ungauged Basins (PUB) (Kratzert et al., 2019a; Yin et al., 2021b),
multistep predictions (Kao et al., 2020; Yin et al., 2021a; Xiang et al., 2020), predictions at multiple timescales (Gauch et al.,
2021a) and regional modelling (Kratzert et al., 2019b; Feng et al., 2020).

The lack of physical mechanism is a critical issue in the use of the LSTM network as it is a black box model (Read et al.,
2019; Reichstein et al., 2019; Xie et al., 2021; Zhao et al., 2019). One the one hand, without explicit physical mechanism such
as the conservation of mass and energy, the LSTM network cannot guarantee causal relationships as physical models can
(Wang et al., 2023; Xie et al., 2021), which may lead to spurious and inaccurate prediction that is potential to violate water
balance, particularly when extrapolating beyond the range of training data (Bhasme et al., 2022; Reichstein et al., 2019). This
property reduces the credibility of the outputs of the LSTM network and limits its application (Cai et al., 2022; Read et al.,
2019; Wang et al., 2023). On the other hand, the lack of physical mechanism leads to the heavy reliance of the LSTM network
on available observations (Read et al., 2019; Xie et al., 2021). Usually, the LSTM network requires a large amount of training
data to learn the dynamics of complex systems so as to achieve robust performance (Gauch et al., 2021b; Kratzert et al., 2019b;
Tsai et al., 2021; Yang et al., 2020).



There is recently a growing attention to the water balance constraint for the LSTM network (Frame et al., 2023; Hoedt et al., 2021; Nearing et al., 2020; Pokharel et al., 2023; Wi and Steinschneider, 2022). It has been found that the water balance constraint can enhance the accuracy and extrapolation ability of the LSTM network (Cai et al., 2022; Wang et al., 2023). Meanwhile, using an architecturally mass-conserving variant of the LSTM (MC-LSTM) (Hoedt et al., 2021), it has recently been observed that the water balance constraint can impair the predictive performance under extreme events (Frame et al.,

2023, 2022). Therefore, there is yet no consensus on the effects of the water balance constraint on the use of the LSTM network. Aiming to bridge the gap, this paper focuses on how the water balance constraint in model architecture affects the robustness of the basin-wise trained LSTM network for rainfall-runoff prediction. The objectives are (1) to investigate the robustness of the LSTM and MC-LSTM against data sparsity, (2) to assess their stability across random parameters initialization and (3) to verify their transferability under contrasting climate conditions. To this end, large-sample tests for rainfall-runoff prediction

are devised based on the Catchment Attributes and Meteorology for Large-sample Studies (CAMELS) dataset across the contiguous United States.

## 2 Methods

### 2.1 LSTM

The LSTM network takes a recurrent architecture, allowing information to be stored and passed over time steps through the cell state vector ($c^t$) and the hidden state vector ($h^t$) (Hochreiter and Schmidhuber, 1997; Jiang et al., 2022). At each time step $t$, the recurrent unit utilizes the current input ($X^t$) and previous hidden state ($h^{t-1}$) to calculate three gates, the input gate ($i^t$), forget gate ($f^t$) and output gate ($o^t$), which control what new information to add in, what previous information to forget and what current information to output, respectively. Finally, the hidden state ($h^t$) is passes through a head layer to derive the

final prediction. The above process can be formulated as follows:

$$\begin{cases} f^t = \sigma\left(W_{xf}X^t + W_{hf}h^{t-1} + b_f\right) \\ \widetilde{c^t} = tanh(W_{xc}X^t + W_{hc}h^{t-1} + b_c) \\ i^t = \sigma(W_{xi}X^t + W_{hi}h^{t-1} + b_i) \\ c^t = f_t \odot c^{t-1} + i^t \odot \widetilde{c^t} \\ o^t = \sigma(W_{xo}X^t + W_{ho}h^{t-1} + b_o) \\ h^t = o^t \odot \tanh(c^t) \end{cases} \quad (1)$$

where $W$ and $b$ respectively indicate learnable weights and bias to be calibrated during training period. Additionally, $\sigma, tanh$ and $\odot$ represent the sigmoid function, the tanh function and the element-wise multiplication, respectively.



## 2.2 Water balance constraint

The Theory-Guided Data Science (TGDS) (Faghmous et al., 2014; Faghmous and Kumar, 2014; Karpatne et al., 2017) has presented a new paradigm to incorporate physical constraints into DL models so that their predictions tend to be physically consistent (Jiang et al., 2020; Karniadakis et al., 2021; Read et al., 2019; Wang et al., 2023; Wi and Steinschneider, 2022). As one of the TGDS strategies, the mass-conserving LSTM (MC-LSTM) is an architecturally mass-conserving variant of the LSTM network (Hoedt et al., 2021). Specifically, the mass conservation constraint is incorporated into the architecture of the LSTM network in order to enforce water balance in rainfall-runoff prediction (Frame et al., 2023, 2022; Hoedt et al., 2021; Nearing et al., 2021).

The MC-LSTM employs normalized activation functions and subtracts the output mass from the storage mass to enforce conservation laws in the architecture of the LSTM network. According to whether directly related to the mass, input variables are distinguished between mass inputs ($x^t$) and auxiliary inputs ($a^t$). The normalized activation functions are used in the input gate ($i^t$) and the forget gate ($R^t$) to guarantee that mass is conserved from the mass inputs ($x^t$) and the previous cell states ($c^{t-1}$). Furthermore, the output mass ($h^t$) is subtracted from the total mass ($m^t$) through the output gate ($o^t$) to keep mass conserved between the cell states ($c^t$) and the output mass. Mathematically, the MC-LSTM is described as follows:

$$i^t = \tilde{\sigma}\left(W_i a^t + U_i \frac{c^{t-1}}{\|c^{t-1}\|_1} + V_i x^t + b_i\right) \tag{2}$$

$$o^t = \sigma\left(W_o a^t + U_o \frac{c^{t-1}}{\|c^{t-1}\|_1} + V_o x^t + b_o\right) \tag{3}$$

$$R^t = \widetilde{ReLU}\left(W_r a^t + U_r \frac{c^{t-1}}{\|c^{t-1}\|_1} + V_r x^t + b_r\right) \tag{4}$$

$$m^t = R^t c^{t-1} + i^t x^t \tag{5}$$

$$c^t = (1 - o^t) \odot m^t \tag{6}$$

$$h^t = o^t \odot m^t \tag{7}$$

$$y_t = \sum_{i=2}^{n} h_i^t \tag{8}$$

where $W$, $U$ and $V$ represent learnable weights; $b$ denotes the learnable bias; $\tilde{\sigma}$ and $\widetilde{ReLU}$ indicate the normalized sigmoid function and the normalized ReLU function as Eq. (9) and Eq. (10), respectively.

$$\tilde{\sigma}(i_k) = \frac{\sigma(i_k)}{\sum_k \sigma(i_k)} \tag{9}$$

$$\widetilde{ReLU}(s_k) = \frac{max(s_k, 0)}{\sum_k max(s_k, 0)} \tag{10}$$



For unobserved mass sinks, e.g., evapotranspiration, the MC-LSTM takes a subset of the output mass vector to accumulate the output water that does not convert to runoff. Given that, the runoff ($y_t$) is the sum of the output mass vector, excluding that subset representing the unobserved mass sinks, shown in equation (8). Accordingly, the internal calculations of the MC-LSTM ensure strictly mass-conservation (here water balance) at any timesteps, between inputs (here precipitation), outputs (here runoff and other sinks) and cell states (here water storage) (Frame et al., 2023).


### 2.3 EXP-HYDRO

The EXP-HYDRO model is employed to benchmark the performances of the LSTM and MC-LSTM. The EXP-HYDRO model is a daily conceptual hydrological model that strictly adheres to mass conservation (Patil and Stieglitz, 2014). It has 2 state variables referred to as a snow accumulation bucket ($S_0$) and a catchment bucket ($S_1$) with the water balance equation

expressed as:

$$\begin{cases} \dfrac{dS_0}{dt} = P_s - M \\ \dfrac{dS_1}{dt} = P_r + M - ET - Q \end{cases} \tag{11}$$

where $M$, $ET$, $Q$, $P_s$ and $P_r$ are 5 flux variables, representing the snowmelt (mm/day), evapotranspiration (mm/day), streamflow (mm/day), daily snowfall (mm/day) and rainfall (mm/day), respectively, calculated by 3 input variables (the daily precipitation ($P$, mm/day), temperature ($T$, ℃) and day length ($L_{day}$, hour)).

In this paper, the EXP-HYDRO model is wrapped with the DL architecture. That is, the non-analytically solvable ordinary differential equations (ODEs) of the EXP-HYDRO model are incorporated into the DL model (Jiang et al., 2020). Therefore, the model parameters are learnable during training period like a DL model while the internal calculations follow the ODEs of the EXP-HYDRO model. Comparing to the MC-LSTM, the EXP-HYDRO model has more specific physical processes to distribute water.


## 3 Large-sample tests

### 3.1 The CAMELS dataset

Large-sample tests are devised based on the CAMELS dataset that comprises daily streamflow observations, catchment attributes and three basin-averaged daily meteorological forcing inputs for 671 basins across the contiguous United States over

the period from1980 to 2010 (Addor et al., 2017; Newman et al., 2015). The CAMELS dataset has been used to support benchmark studies, generalization and application to other scenarios for its sufficiently long hydrometeorological time series and a large number of diverse basins (Feng et al., 2020; Frame et al., 2022; Kratzert et al., 2021, 2019b; Yin et al., 2021a).



The three meteorological forcing data in the CAMELS dataset are derived from three different gridded data products, i.e., Daymet, Maurer and North American Land Data Assimilation System (NLDAS). The Daymet (Thornton et al., 1997) is chosen as the forcing inputs here because of its high spatial resolution (1 km × 1km) and better forcing quality (Feng et al., 2022; Newman et al., 2015). For a direct comparison with the previous studies mentioned above, 531 basins are used, while other basins with an area greater than 2000 km$^2$ or showing large discrepancies in their areas when calculated using different strategies are removed. In the LSTM, MC-LSTM and EXP-HYDRO modelling, the runoff is taken as the target variable and the precipitation, temperature, vapor pressure, solar radiation and day length as forcing variables.

## 3.2 Experimental design

Four experiments are set up to assess the effects of the water balance constraint on the robustness of the basin-wise trained LSTM for rainfall-runoff prediction in different aspects (Fig. 1). For each experiment, three types of models with different degrees of physical constraints, a standard LSTM, a MC-LSTM and a DL wrapped EXP-HYDRO, are trained independently for each catchment. Experiment 1 tests the basic performances of the LSTM, MC-LSTM and EXP-HYDRO models in rainfall-runoff prediction for 531 basins. Experiment 2 aims to determine how the water balance constraint affects the sensitivity of the LSTM network to data sparsity. Based on the predictions from Experiment 1 and Experiment 2, Experiment 3 quantifies the impact of the water balance constraint on the stability of the LSTM network against random parameters initialization. Experiment 4 is designed to assess the transferability of the three models under contrasting climate conditions and verify whether the water balance constraint enhances the transferability of the LSTM network. According to the results of Experiment 2, the training period length of Experiment 4 is set to be 6 years, which is considered to be sufficient to ensure transferability yet provide robust training.



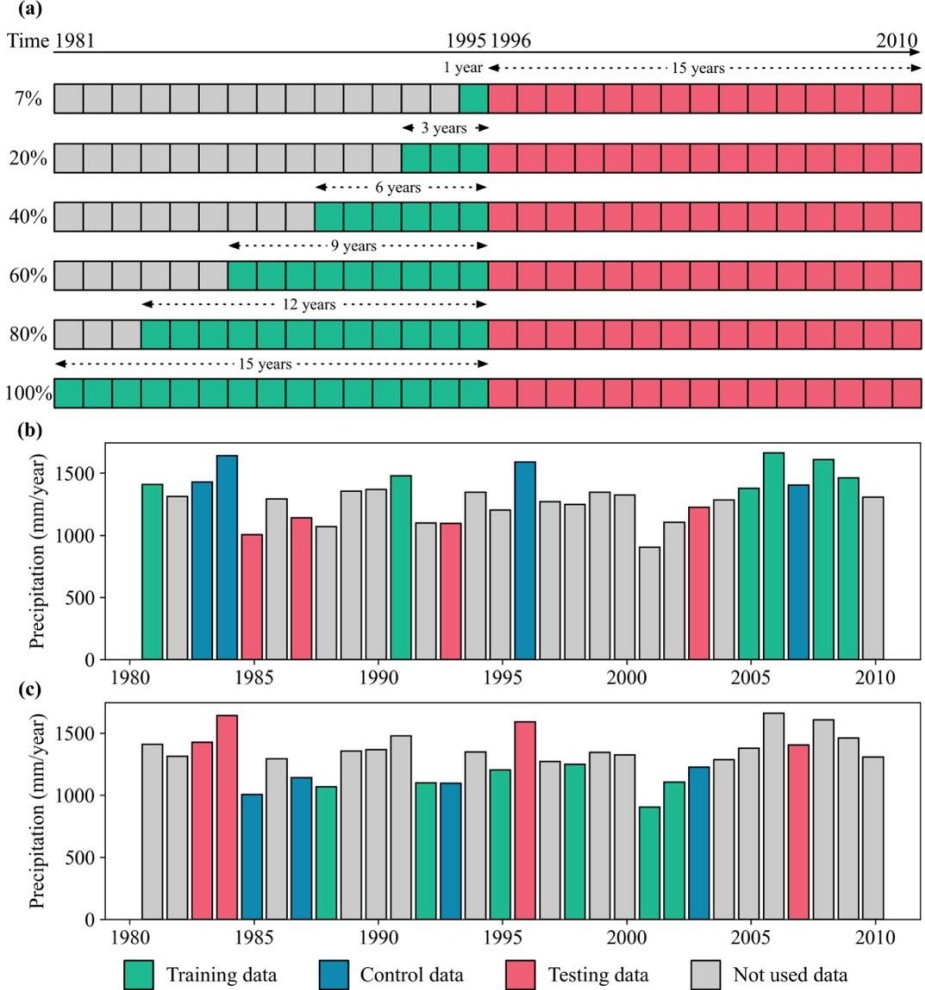

**Figure 1.** Schematic of (a) the split-sample for different sparse training datasets at each catchment and (b) (c) the modified DSST at catchment 01022500 (USGS code) based on annual precipitation for hydrological year (1 October to 30 September).

Experiment 1: Performances of LSTM, MC-LSTM and EXP-HYDRO

In order to test the predictive performances of the LSTM, MC-LSTM and EXP-HYDRO in modelling rainfall-runoff processes, these models are trained separately for each of the 531 basins. The performances of the LSTM and MC-LSTM are compared to quantify the general impact of the water balance constraint in model architecture to the LSTM network, while the EXP-HYDRO model serves as a benchmark. For each catchment, the training period covers 15 water years (from 1 October 1980 to 30 September 1995) and the testing period also covers 15 water years (from 1 October 1995 to 30 September 2010). Given the uncertainty caused by the random initialization of model parameters, the training and testing of each model are



repeated for 10 times with different random seeds. In total, Experiment 1 yields 15,930 models, 30 models for each of the 531 catchments.

       Experiment 2: Sensitivity to data sparsity

       To determine how the water balance constraint affects the sensitivity of the LSTM network to data sparsity, the LSTM, MC-LSTM and EXP-HYDRO models are trained on various sparse training datasets and then evaluated on the same testing

datasets. For each catchment, the training period and testing period in Experiment 1 are considered as the complete training period and the testing period, respectively. Sparse training datasets are constructed for each basin by continually removing the data of entire water years from the earlier years of the whole training period instead of randomly removing, which can avoid excessive destruction of time dependency between data and simulate real scenarios that lack historical data (Read et al., 2019). Therefore, the sparse training datasets are set to 7%, 20%, 40%, 60% and 80% of the whole training data with the training

period lengths ranging from 1 year to 12 years, as shown in Fig. 1a. Trained and tested repeatedly 10 times with different random seeds over the 531 basins, a total of 79,650 ($10 \times 3 \times 5 \times 531$) models are obtained.

       Experiment 3: Stability against random parameters initialization

       Owing to the stochastic nature of the training process, the LSTM network can be disturbed by the initialization of model parameters (Kratzert et al., 2018). Accordingly, it is helpful to independently repeat the training and testing procedures for

several times with different random seeds in order to eliminate the uncertainty caused by random initialization of model parameters (Feng et al., 2020; Kratzert et al., 2019a). While the average performance of the ensemble models is then considered as the stable performance, the differences among the performances of models with different random seeds reflect the stability of the specific model against random parameters initialization. Experiment 3 calculates the standard deviation of the performances from the 10-member ensemble models in Experiment 1 and Experiment 2 as the stability of the model.

190       Experiment 4: Transferability under contrasting climate conditions

       The transferability is examined through a modified version (Broderick et al., 2016) of differential split sample testing (DSST) (KLEMEŠ, 1986). There are three datasets, the training, the control (independent but similar to the training data) and the testing data (independent and opposite to the training and the control data). Models are trained based on the training data, and the differences in performances between the control (in-bound test) and the testing data (out-of-band test) are indicative

of transferability (Broderick et al., 2016). For each catchment, two scenarios are conducted to examine the transferability between the wettest and the driest hydrological years that identified from the total precipitation of hydrological years, while W/D (D/W) represents the scenario of training on the wettest (driest) years and then testing on the driest (wettest) years. In the W/D scenario, each model is trained using the first, third, fifth, sixth, eighth and tenth ranked wettest years, while model performance on the second, fourth, seventh and ninth ranked wettest years provides a benchmark to assess the transferability

of the model tested on the contrasting second, fourth, seventh and ninth ranked driest years (as an example of the basin 01022500 shown in Fig. 1b). In the D/W scenario, the transferability assessment is conducted using the opposite driest and





wettest years (as an example shown in Fig. 1c). Removing catchments without complete data from 1981 to 2010, 475 basins remain here, thus a total of 28,500 ($10 \times 3 \times 2 \times 475$) models are trained and tested.

## 3.3 Model training and evaluation

All input variables of the LSTM are normalized by removing the mean and scaling by the standard deviation. For the MC-LSTM, the auxiliary inputs (input variables excluding precipitation) are normalized while the mass input (precipitation) not. The MC-LSTM is architecturally constrained to the water balance so that they do not utilize dropout strategy. In order to compromise between maximumly reducing the uncertainty caused by different numbers of model parameters and achieving potentially more powerful predictions, the hidden sizes of the LSTM and MC-LSTM networks are set to 50 and 20, respectively, so that their numbers of parameters differ by less than 0.1%. As the EXP-HYDRO model is a process-based model, there is no need for the DL wrapped EXP-HYDRO model to normalize their input variables and to set the hidden size or dropout rate. Excluding these settings above, the LSTM, MC-LSTM and EXP-HYDRO models in the four experiments have the same hyperparameters (shown in Table 1) and the same loss function:

$$FVU = \frac{\sum_{n=1}^{N}(y_n - \hat{y}_n)^2}{\sum_{n=1}^{N}(y_n - \bar{y})^2} \tag{12}$$

where N is the number of samples; $\hat{y}$ and $y_n$ represent the simulated runoff and its corresponding observation, respectively; $\bar{y}$ is the averaged value of observed runoff.

**Table 1.** Hyperparameters of the LSTM, MC-LSTM and EXP-HYDRO models

| Hyperparameter | LSTM | MC-LSTM | EXP-HYDRO |
| --- | --- | --- | --- |
| Batch size | 256 | 256 | 256 |
| Initial learning rate | 0.01 | 0.01 | 0.01 |
| Learning rate decay | 0.3 | 0.3 | 0.3 |
| Input time step (day) | 365 | 365 | 365 |
| Lead time (day) | 1 | 1 | 1 |
| Hidden size | 50 | 20 | - |
| Dropout rate | 0.4 | - | - |
| Epoch | Early stop | Early stop | Early stop |
| Optimizer | Adam | Adam | Adam |




The Kling-Gupta Efficiency (KGE) (Gupta et al., 2009) is used to quantify the performances of the rainfall-runoff predictions here. As shown below, KGE summarizes model performance in three key aspects: correlation, bias and variance.

$$KGE = 1 - \sqrt{(r-1)^2 + (\beta - 1)^2 + (\gamma - 1)^2} \tag{13}$$

$$\beta = \frac{\mu_{sim}}{\mu_{obs}} \tag{14}$$

$$\gamma = \frac{\sigma_{sim}}{\sigma_{obs}} \tag{15}$$


where $r$ is the correlation coefficient between simulations and observations; $\beta$ represents the ratio between mean simulations and mean observations; $\gamma$ measures the relative variability in the simulations and observations; $\mu$ and $\sigma$ represent the mean and standard deviation of the runoff series, respectively (Gupta et al., 2009).The value of KGE varies from negative infinity to 1, with a value closer to 1 suggesting superior model performance.

The model performance comprises two essential aspects: accuracy and robustness. The accuracy is calculated as the mean KGE of the 10-member ensembles. The robustness is composed of three aspects: (a) the sensitivity to data sparsity, (b) the stability against random parameters initialization and (c) the transferability under contrasting climate conditions. In experiment 2, the sensitivity of a model at a catchment is estimated from the variation range in the KGE under varying sparse training data sets. A larger variation range suggests the high sensitivity of a model to data sparsity. In experiment 3, the stability of a model

is calculated as the standard deviation of KGE values from 10-member ensembles, while a higher standard deviation of KGE values indicates the worse stability of a model across random parameters initialization. In experiment 4, the transferability is estimated as the KGE difference between the control data and the corresponding testing data. The transferability of a model is better with a lower KGE difference under contrasting climate conditions.

## 4 Results

### 4.1 Performances of LSTM, MC-LSTM and EXP-HYDRO

The predictive performances of the LSTM, MC-LSTM and EXP-HYDRO models across the 531 catchments are examined by the KGE in Fig. 2. Specifically, the left panel illustrates the boxplot of the KGE values and the right panel displays the empirical cumulative distributions. It can be observed that there is marginal difference in the performances of the LSTM

and MC-LSTM as the median KGE is respectively 0.676 and 0.679 (shown in the penultimate column in Table 2). In particular, the distributions of mean KGE for the LSTM and MC-LSTM are similar, which is illustrated by their cumulative distribution curves that are close to overlapping in Fig. 2. These results indicate that the water balance constraint has little impact on the general accuracy of the LSTM network when it is trained with data of 15 years for a single basin, which is consistent with



previous studies (Hoedt et al., 2021; Nearing et al., 2020). In the meantime, the LSTM and MC-LSTM generally outperform
the EXP-HYDRO model, which is visually shown by the rightward shifts of the empirical cumulative distributions for the
LSTM and MC-LSTM compared to the EXP-HYDRO models in Fig. 2.

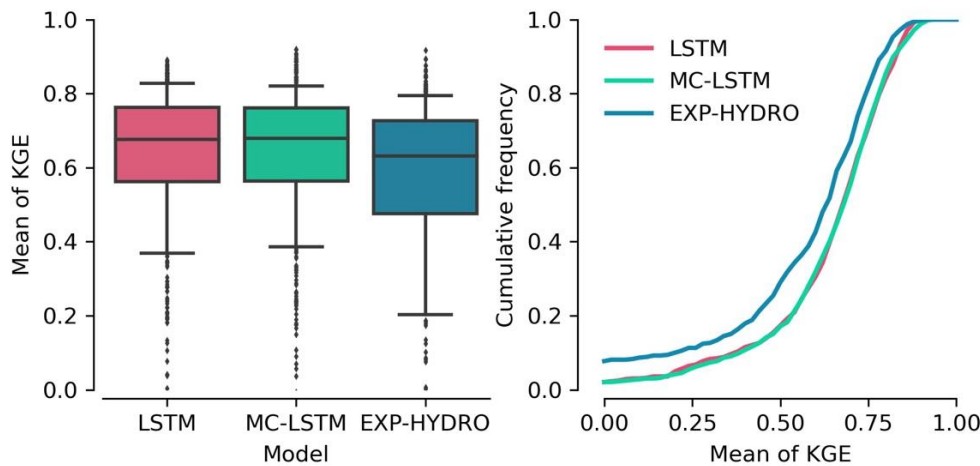

**Figure 2.** Empirical cumulative distributions and boxplot of the evaluation performances for the LSTM, MC-LSTM and EXP-HYDRO
models across the 531 catchments. The performance of each model for each basin is summarized by the mean value of KGE across different
random seeds.

## 4.2 Sensitivity to data sparsity

In the testing period, the hydrographs of two basins where different models are trained with different amounts of training
data are shown in Fig. 3. Not surprisingly, the LSTM network exhibits the largest variability of KGE values when the training
period length is 3 years. As the length of training data is increased from 3 years to 15 years, the variability of ensemble KGE
values for the LSTM decreases, while there are less changes of the variability of the ensemble KGE values for the MC-LSTM
and EXP-HYDRO models. In the basin 01439500, the MC-LSTM and EXP-HYDRO models outperform the LSTM with
better goodness of fit and more narrow ranges of predictions. With the length of training data increased from 3 years to 15
years, the LSTM becomes more accurate, and its prediction ranges narrow. In the basin 11532500, the LSTM, MC-LSTM and
EXP-HYDRO models fit well with both 3- and 15-year training data. The LSTM network becomes more accurate with more
training data.



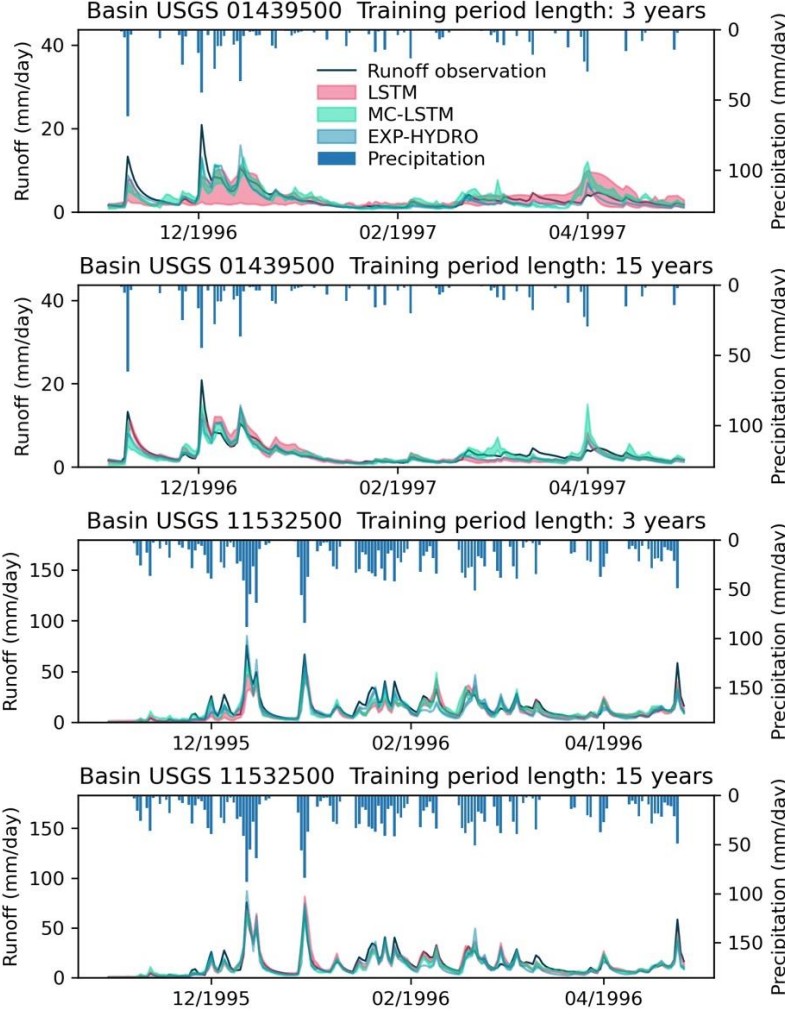

**Figure 3.** Hydrograph comparisons between different models with two different training period lengths. The bands of runoff represent the prediction ranges of 10-member ensembles with different random seeds.

Figure 4 presents an illustration of the KGE values across the 531 catchments under different data sparsity. It can be observed that the accuracy of the three models is affected by data sparsity and that the extent of the impacts varies. Specifically, as the amount of training data increases from 1 year to 15 years, the LSTM network benefits the most with median KGE increased by 95.0% (shown in Table 2). By contrast, the MC-LSTM network is less affected and the median KGE is increased by 32.7%. Additionally, the accuracy of the LSTM rapidly increases by 0.459 (from 0.034 to 0.493) with the training data length increased from 1 to 3 years, while less than 0.123 (median KGE from 0.457 to 0.580) for the MC-LSTM. These results suggest that the incorporation of the water balance constraint reduces the sensitivity of the LSTM network to data sparsity,





which is consistent with conclusions of the previous study that physical constraints reduce the data volume dependency of the
LSTM (Read et al., 2019; Wang et al., 2023).

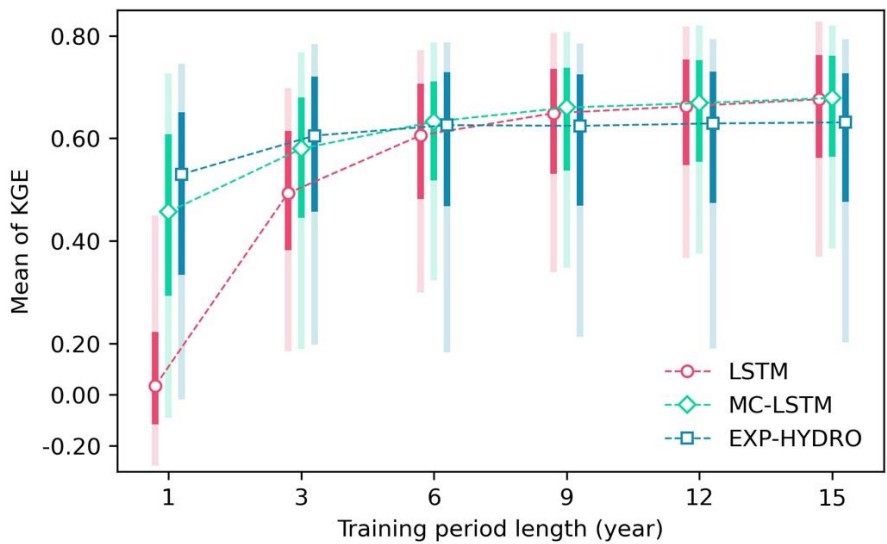

**Figure 4.** Ranges of the mean values of KGE for the LSTM, MC-LSTM and EXP-HYDRO models with different data sparsity across the
531 catchments. The markers, deep-colour bars and light-colour bars respectively represent the median, [25%, 75%] and [10%, 90%] inter-
quantile ranges over the 531 catchments.

**Table 2.** Median values of mean KGE for the LSTM, MC-LSTM and EXP-HYDRO models with different data sparsity across the 531
catchments.

| Training period (year) | 1 | 3 | 6 | 9 | 12 | 15 | Max Δ (%) |
|---|---|---|---|---|---|---|---|
| LSTM | 0.034 | 0.493 | 0.606 | 0.649 | 0.663 | 0.676 | 95.0% |
| MC-LSTM | 0.457 | 0.580 | 0.633 | 0.660 | 0.669 | 0.679 | 32.7% |
| EXP-HYDRO | 0.530 | 0.605 | 0.626 | 0.624 | 0.629 | 0.631 | 16.0% |

*Note: Max Δ denotes the percentage change of median performance with training period length increased from 1 year to 15 years when the
performances of the models trained with data of 15 years is considered as the complete performance.*





### 4.3 Stability against random parameters initialization

Across the 531 basins, the stability of the LSTM, MC-LSTM and EXP-HYDRO models with different extents of data sparsity is summarized in Fig. 5 and Table 3. Overall, the MC-LSTM has lower standard deviation values of KGE than the LSTM for all sparsity levels of training data. This result suggests that adding the water balance constraint enhances the stability of the LSTM network against random parameters initialization. Specifically, the MC-LSTM, compared to the LSTM, experiences a reduction ranging from 69.5% to 24.0% in the median of the standard deviation of KGE, as shown in Table 3. In addition, the standard deviation values of KGE for the LSTM and MC-LSTM reduce with the training data increased, while 300 the degree of reduction varies for each of the three models. The LSTM exhibits the largest magnitude of reduction in the standard deviation of KGE with the length of training data increased from 1 year to 15 years, while the MC-LSTM and EXP-HYDRO models exhibit slighter reductions. These results also indicate that adding the water balance constraint reduces the sensitivity of the LSTM network to data sparsity.

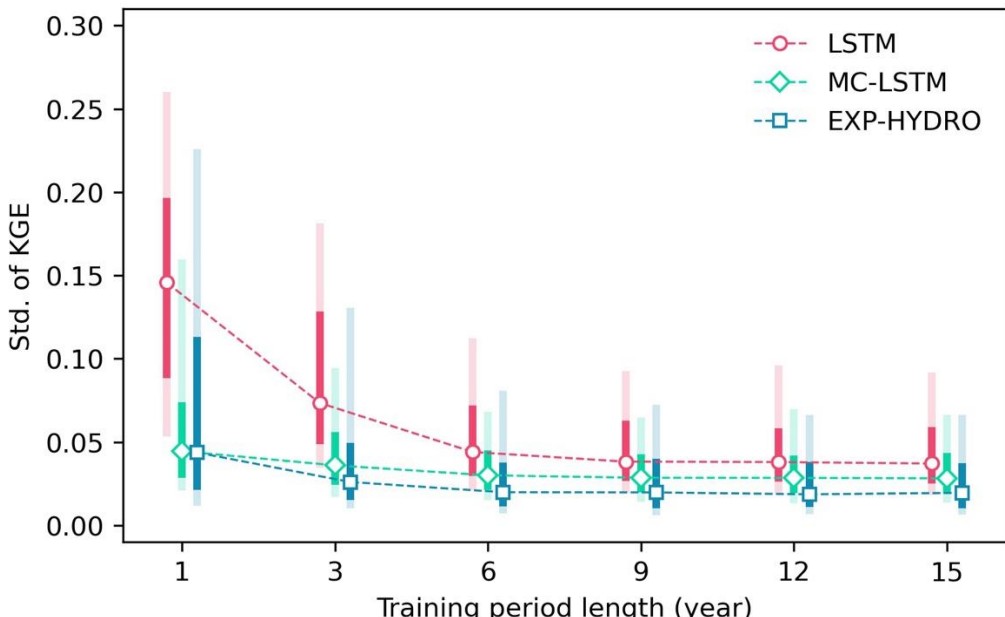


**Figure 5.** Ranges of the standard deviation values of KGE for the LSTM, MC-LSTM and EXP-HYDRO models across the 531 catchments under different data sparsity. The markers, deep-colour bars and light-colour bars respectively represent the median, [25%, 75%] and [10%, 90%] inter-quantile ranges over the 531 catchments.

**Table 3.** Median values of standard deviation of KGE for the LSTM, MC-LSTM and EXP-HYDRO models with different data sparsity across the 531 catchments.




| Training period (year) | 1 | 3 | 6 | 9 | 12 | 15 | Max Δ (%) |
|---|---|---|---|---|---|---|---|
| LSTM | 0.146 | 0.073 | 0.044 | 0.038 | 0.038 | 0.037 | -74.5% |
| MC-LSTM | 0.044 | 0.036 | 0.030 | 0.029 | 0.028 | 0.028 | -36.6% |
| EXP-HYDRO | 0.044 | 0.026 | 0.020 | 0.020 | 0.019 | 0.020 | -55.3% |
| MC Δ (%) | -69.5% | -51.0% | -31.8% | -25.4% | -25.1% | -24.0% | - |

*Note: MC Δdenotes the percentage difference in the median of the standard deviation of KGE between the MC-LSTM and LSTM. Max Δdenotes the percentage change of the standard deviation of KGE with training period length increased from 1 year to 15 years.*


Figure 6 shows the change of the stability at individual basins for the LSTM and MC-LSTM. The MC-LSTM exhibits remarkably smaller standard deviation of KGE than the LSTM. Specifically, the MC-LSTM tends to be more stable at a total of 450 (85%), 386 (73%) and 366 (69%) basins when models are trained with data of 3 years, 9 years and 15 years, respectively. These results illustrate that the water balance constraint improves the stability of the LSTM network. In addition, the number of basins where the MC-LSTM is more stable than the LSTM has reduced with the training period length increased from 1 year to 15 years, and the differences (as shown in Table 3) in the median of the KGE standard deviation values between the MC-LSTM and LSTM decrease from -69.5% to -24.0%. The implication is that increasing the training data can narrow the stability differences between the LSTM and MC-LSTM, compensating for the instability of the LSTM network caused by the lack of the water balance constraint.




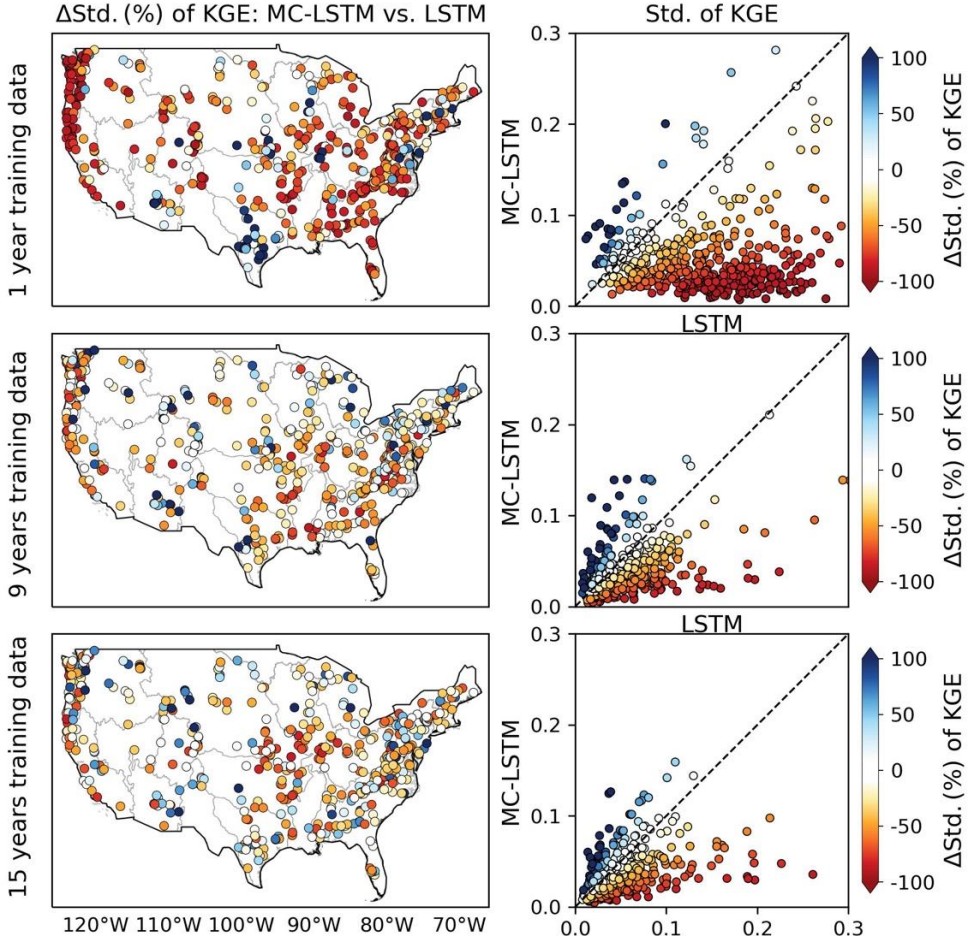

**Figure 6.** Per-basin change of stability between the MC-LSTM and LSTM in the 531 basins. Red dots indicate basins where adding the water balance constraint improves the stability of the LSTM (darker indicates larger relative improvement), and blue dots indicate basins where there is a decrease in stability (darker indicates worse relative detriment). Note that a higher standard deviation (Std.) of KGE indicates
the worse stability of a model across random parameters initialization.

## 4.4 Transferability under contrasting climate conditions

The KGE values for all models under the control and testing conditions across the 475 catchments are shown in Fig. 7 to examine the transferability between the wettest/driest years. In general, the MC-LSTM exhibits higher transferability than the
LSTM. This result suggests that adding the water balance constraint advances the transferability of the LSTM under contrasting climate conditions. In both W/D and D/W scenarios, the LSTM, MC-LSTM and EXP-HYDRO models illustrate close accuracy on the control data with KGE range of [0.634, 0.664] and [0.598, 0.602], as demonstrated in Table 4. When



transferring to the testing data, the MC-LSTM outperforms the LSTM with median KGE of 0.576 in the W/D scenario and 0.515 in the D/W scenario, compared to the median KGE of 0.538 and 0.446 for the LSTM, respectively. In terms of median

KGE differences between the control and testing data, the MC-LSTM exhibits less degradation in performance than the LSTM with median KGE decreased by 13.3% in the W/D scenario and 15.4% in the D/W scenario, compared to the median KGE decreased by 17.4% and 25.4% for the LSTM, respectively.

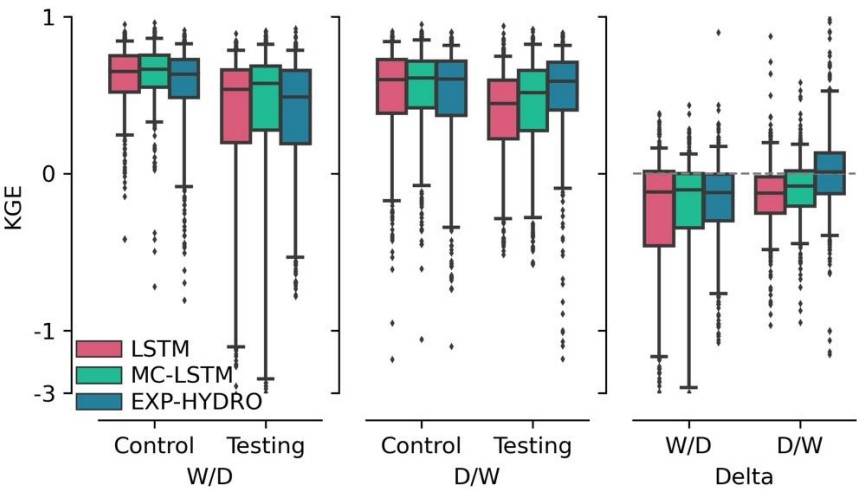

**Figure 7.** Ranges of KGE of the LSTM, MC-LSTM and EXP-HYDRO models across the 475 catchments for different DSST scenarios. Delta denotes the KGE difference between the testing data (Testing) and control data (Control).

**Table 4.** Median KGE and median KGE differences (percent) between the control data and the testing data for different models in different DSST scenarios across the 475 basins.

| Model | W/D | | | D/W | | |
|---|---|---|---|---|---|---|
| | Control | Testing | Δ (%) | Control | Testing | Δ (%) |
| LSTM | 0.651 | 0.538 | -17.4 | 0.598 | 0.446 | -25.4 |
| MC-LSTM | 0.664 | 0.576 | -13.3 | 0.609 | 0.515 | -15.4 |
| EXP-HYDRO | 0.634 | 0.489 | -23.0 | 0.602 | 0.590 | -2.2 |

*Note: Δ denotes the median of the KGE differences between the control data and testing data.*

The per-basin accuracy comparisons between the LSTM and MC-LSTM in the testing conditions are plotted in Fig. 8. In the D/W scenario, the MC-LSTM exhibits higher KGE values compared to the LSTM across 318 basins (67%). But for the



W/D scenario, the number of basins with higher KGE for the MC-LSTM than the LSTM decreases to 262 (55%). The

catchments where the MC-LSTM shows improved accuracy than the LSTM are mainly located in the central arid regions of

the United States. In these areas, the runoff generation is dominated by the infiltration-excess overland flow, which is largely

controlled by short-duration, high-intensity precipitation events (Berghuijs et al., 2016). These results suggest that the

consideration of the water balance constraint improves the prediction of the LSTM transferred from the driest to the wettest

years. For the prediction transferred from the wettest to the driest years, the LSTM itself demonstrates good accuracy and the

improvement from enforcing the water balance constraint is not substantial. It is possibly due to the skewness and censoring

characteristics of hydroclimatic variables (Huang et al., 2023), with lower runoff values occurring much more frequently than

higher values. This property makes the training data of the wettest years more similar to the testing data of the driest years,

thereby making it easier for the LSTM to transfer in the W/D scenario.

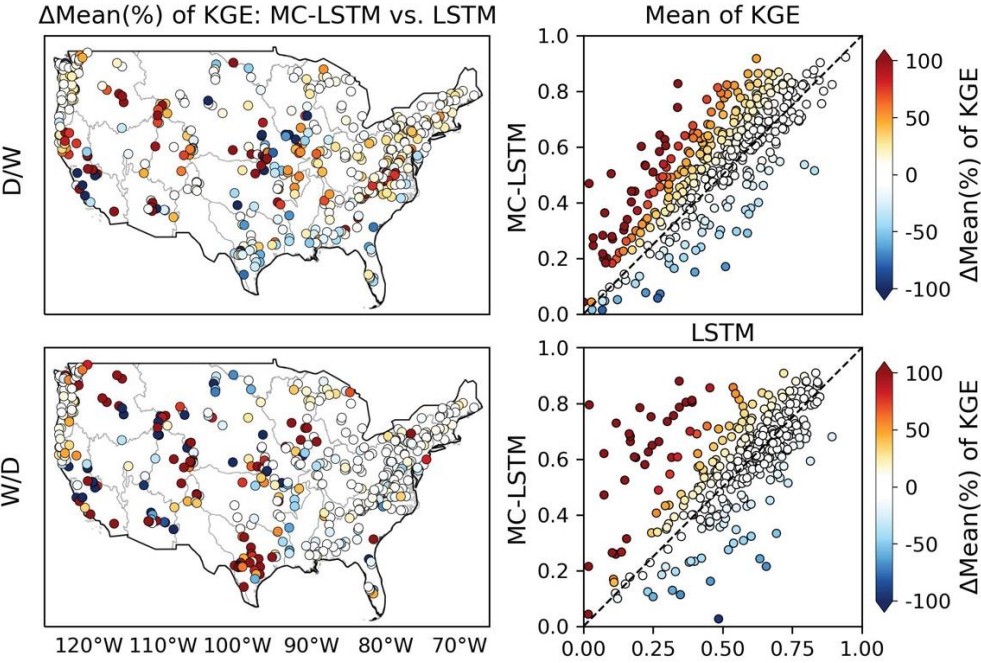


**Figure 8.** Per-basin change of accuracy between the MC-LSTM and LSTM at the 475 basins. Note that higher accuracy indicates higher

transferability in testing conditions. Red dots indicate basins where adding the water balance constraint improved the transferability over the

LSTM (darker indicates larger relative improvement), and blue dots indicate basins where the transferability decreases (darker indicates

worse relative detriment).




## 5 Discussion

It has been highlighted that more training data contributes to the performance of the LSTM network (Gauch et al., 2021b; Read et al., 2019; Wang et al., 2023; Xie et al., 2021). In this paper, the role of the training data in the performance of the MC-LSTM network is investigated through large-sample tests. The findings generally conform to previous findings that the LSTM
trained to hundreds of basins allows for better streamflow predictions in a given basin compared to the LSTM only trained to that specific basin or a smaller subset of basins (Jiang et al., 2020; Kratzert et al., 2019b), even for out-of-sample predictions (Gauch et al., 2021b; Xie et al., 2021). These results suggest that DL models can gain extrapolation ability from big data compensating the lack of physical mechanisms (Feng et al., 2021; Kratzert et al., 2019b; Nearing et al., 2019). Through extensive training with large amounts of data, DL models can advance hydrologic predictions, even without explicit physical
mechanisms (Nearing et al., 2021; Wi and Steinschneider, 2022). Though experiments here do not evaluate the impact of data quality or uncertainty in model inputs on the quality of predictions, such efforts could provide additional insights beyond the scope of this paper (Read et al., 2019).

Given that data is not always sufficient, the sensitivity of DL models when given scarce training data is essentially important (Feng et al., 2021; Gauch et al., 2021b). The TGDS provides effective tools for reducing data requirements of DL
models (Karniadakis et al., 2021; Karpatne et al., 2017; Read et al., 2019; Xie et al., 2021). In this paper, the incorporation of the water balance constraint into the LSTM network provides direct guidance based on physical knowledge to the internal calculation processes, thus reducing the need to learn this specific physical mechanism form large amounts of data (Frame et al., 2023; Hoedt et al., 2021). Another TGDS strategy, which reconfigures the loss functions with physical penalties, allows DL models to obtain additional guidance to their training process from the physical mechanisms, also leading to less training
data requirement (Wang et al., 2023; Yang et al., 2020; Zhao et al., 2019). Moreover, recent studies showed that using synthetic data generated by physical models to pretrain DL models could also contribute to overcome the conditions of sparse observation data (Read et al., 2019; Xie et al., 2021; Zhang et al., 2022). Furthermore, this paper also demonstrates the potential of enforcing physical constraints such as the water balance constraint to strengthen the robustness of DL models under random parameters initialization and climate change.

Although TGDS models can provide more accurate and robust predictions than pure DL models in basin-wise scale or data scarce conditions, it deserves additional scrutiny when trained with data from a large number of diverse basins (Frame et al., 2022; Nearing et al., 2021; Wi and Steinschneider, 2022). Recent studies have illustrated that for DL models, physical constraints are effective in local models but offer little improvement in the regional models (Frame et al., 2023; Xie et al., 2021), even reduce predictive performance under extreme events (Frame et al., 2022). This outcome can be attributed to that
catchments with similar flood generating processes have some similar outliers (Bertola et al., 2023) and that extreme events that did not occur frequently in one basin may occur in other basins (Xie et al., 2021). Therefore, there seems to be a compensating effect between data and knowledge on DL models, where the process knowledge is crucial for models trained with sparse data but less important with sufficient data. With much more available data to learn the patterns that hydrologic



systems respond to previously unobserved extreme events and climate conditions, large-sample hydrology is expected to

enhance the performances of DL models for extreme events predictions and climate change projections (Bertola et al., 2023; Wi and Steinschneider, 2022).

## 6 Conclusions

This paper is concentrated on the effects of the water balance constraint in model architecture on the robustness of the

basin-wise trained LSTM for rainfall-runoff prediction. That is, large-sample tests based on CAMELS dataset are conducted to assess the robustness of the LSTM and its architecturally mass-conserving variant (MC-LSTM) from three perspectives, i.e., the sensitivity to data sparsity, the stability across random parameters initialization and the transferability under contrasting climate conditions. The results show that the water balance constraint contributes to the robustness of the basin-wise trained LSTM. One finding is that for varying data sparsity of training data ranging from 1 year to 15 years, the addition of the water

balance constraint decreases the sensitivity of the LSTM to data sparsity from 95.0% to 32.7%. Another finding is that the water balance constraint is effective in improving the stability of the LSTM at 450 (85%) when available data are 3 years. The third finding is that the water balance constraint enhances the transferability of the LSTM from the driest years to the wettest years at 318 (67%) basins. The in-depth investigations of this paper facilitate insights into the use of the LSTM network and other DL models for rainfall-runoff modelling.


*Data availability*. The CAMELS dataset is download from https://gdex.ucar.edu/dataset/camels.html (Addor et al., 2017).

*Author contribution*. TZ and QL designed the experiments. QL carried them out. QL and TZ developed the model code and performed the experiments. TZ and QL prepared the manuscript with contributions from the co-authors.

*Competing interests*. The contact author has declared that none of the authors has any competing interests.

*Acknowledgements*. This research is supported by the National Natural Science Foundation of China (U1911204, 51725905, 52130907, 51979295, 51861125203 and 52109046), the National Key Research and Development Program of China (2021YFC3001000) and the Guangdong Provincial Department of Science and Technology (2019ZT08G090).



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
