# Peer review of "Role of the water balance constraint in the long short-term memory network: large-sample tests of rainfall-runoff prediction"

_EGUsphere, 2023_

## Author Comment (AC1)

**Response**

*Anonymous Referee #1:*

*In this paper, four experiments were conducted to assess the performance of three types of Rainfall-Runoff models: Long Short-Term Memory (LSTM), mass conservative LSTM (MC-LSTM), and the conceptual model EXP-HYDRO. The document is well-organized, and the results are presented and discussed clearly.*

We are grateful to you for the positive comments.

*Nevertheless, certain critical aspects arise in the formulation and execution of the study:*

Thank you very much for the constructive comments. We have improved the paper accordingly and provide point-by-point responses.

1.  *The primary justification for the study is based on the authors' assertion that "there is yet no consensus on the effects of the water balance constraint on the use of the LSTM network." To support this claim, the authors cite the works of Cai et al. (2022) and Wang et al. (2023), stating higher accuracies with MC-LSTM over LSTM. However, Cai et al. (2022) does not use LSTM neural networks, and Wang et al. (2023) applies physically informed LSTMs for a different purpose and at a scale unrelated to the Rainfall-Runoff models of Frame et al. (2022) and Frame et al. (2023). Consequently, the comparison appears unfair, involving different processes, evaluated at different scales, and with significantly different instances.*

Thank you for the insightful comment. We are sorry for the confusing information and have modified the expression and citation. With similar scales, same processes and comparable instances, Nearing et al. (2020), Wi and Steinschneider (2024) and Frame et al. (2023, 2022) provide fair comparison for rainfall-runoff prediction in the revision. In the meantime, the focus of this paper is emphasized. The revised text is as follows:

"Using an architecturally mass-conserving variant of the LSTM (MC-LSTM) (Hoedt et al., 2021), it has been found that the water balance constraint can enhance the accuracy and extrapolation ability of the LSTM network for rainfall-runoff prediction (Nearing et al., 2020; Wi and Steinschneider, 2024). On the other hand, it has recently been observed that the water balance constraint can impair the predictive performance under extreme events (Frame et al., 2023, 2022), as well as hard constrained models in multitask hydrological forecasting (Li et al., 2024). Therefore, there is yet no consensus on the effects of the water balance constraint on the use of the LSTM network for rainfall-runoff prediction (Pokharel et al., 2023), particularly its robustness—the ability to perform consistently across varying conditions (Manure et al., 2023)" (Page 3, Lines 55 to 62)

2. *Considering the preceding point, the objective and main conclusion of the work are not clear. While each of the four experiments is thoroughly described, there appears to be a lack of novelty. For instance, the results of experiments 1 and 2 are somewhat expected and have already been published by other authors using diverse datasets.*

Thank you for the constructive comments. We have rewritten the Discussion section to emphasize the significance of this paper:

"It has been highlighted that more training data contributes to the performance of the LSTM network (Gauch et al., 2021b; Read et al., 2019; Wang et al., 2023; Xie et al., 2021). In the case of different training data amounts, the role of the water balance constraint in the performance of the LSTM network is investigated through large-sample tests in different aspects. For single-basin trained DL and TGDS models, previous studies have quantified how additional training data improve their predictions in limited areas (Read et al., 2019; Wang et al., 2023), rather than large-sample tests which can help to understand model limitations and draw robust conclusions from a big-picture perspective (Addor et al., 2020; Gupta et al., 2014). In addition, a modified

DSST method is utilized to assess the transferability of the MC-LSTM and LSTM under contrasting climate conditions in this paper. Recent studies have assessed the credibility of future streamflow projections under warming through metamorphic testing (Razavi, 2021; Reichert et al., 2023; Wi and Steinschneider, 2022, 2024; Yang and Chui, 2021), while metamorphic testing requires partly subjective expert judgements and utilizes no realistic climate change scenarios (Reichert et al., 2023), thereby may leading to unreliable results (Wi and Steinschneider, 2024)." (Pages 18 to 19, Lines 364 to 374)

3. *As highlighted by Kratzert et al. (2024, https://doi.org/10.5194/hess-2023-275), the training of effective rainfall-runoff LSTM models requires the use of multiple basins. However, in this study, several LSTM models are trained with single-basin data. Consequently, suboptimal LSTM models are likely obtained, and the superior performance demonstrated by MC-LSTM may be a consequence of these suboptimal models.*

Thank you for the insightful comments. To clarify the differences and connections between this paper with the mentioned literature above, we have rewritten the Discussion section and the related part are as follows:

"Although TGDS models can provide more accurate and robust predictions than pure DL models in basin-wise scale or data scarce conditions, it deserves additional scrutiny when trained with data from a large number of diverse basins (Frame et al., 2022; Nearing et al., 2021; Wi and Steinschneider, 2022). Recent studies have illustrated that the LSTM network works better for rainfall-runoff prediction when trained with a large amount of hydrologically diverse data than with data from a single watershed (Kratzert et al., 2024). Specifically, for DL models, physical constraints are effective in local models but offer little improvement in the regional models (Frame et al., 2023; Xie et al., 2021), even reduce predictive performance under extreme events (Frame et al., 2022). This outcome can be attributed to that pure DL models might be flexible enough to capture the behaviour in observation data with inconsistent water balance closure

better than DL models constrained by the strict water balance (Kratzert et al., 2024; Frame et al., 2023; Beven, 2020). Besides, catchments with similar flood generating processes and similar characteristics may have some similar outliers and DL models can capture the rainfall-runoff responses among these basins (Xie et al., 2021; Bertola et al., 2023; Wi and Steinschneider, 2024). Therefore, there seems to be a compensating effect between data and knowledge on DL models, where the process knowledge is crucial for models trained with sparse data but less important with sufficient data. Large-sample hydrology is thus expected to enhance the performances of DL models for extreme events predictions and climate change projections (Bertola et al., 2023; Wi and Steinschneider, 2022, 2024).

Given that data is not always sufficient, the sensitivity of DL models when given scarce training data is essentially important (Feng et al., 2021; Gauch et al., 2021b). The TGDS provides effective tools for reducing data requirements of DL models (Karniadakis et al., 2021; Karpatne et al., 2017; Read et al., 2019; Xie et al., 2021). Therefore, this paper explores the effects of the water balance constraint on the robustness of the LSTM under restricted conditions, thereby training single model for each single basin rather than simultaneously for a large number of basins. Although the latter can achieve better performance (Kratzert et al., 2024), it is beyond the scope of this paper but worthy of further study…" (Page 19, Lines 378 to 397)

4. *It is unclear why the authors opted for the hyperparameters listed in Table 1. Moreover, it is essential to consider how the results of their experiments might be influenced by alternative hyperparameters. Could the enhanced performance of MC-LSTM be subject to change with different values of model hyperparameters?*

Thank you for the constructive comments. We are sorry for the unclear information and have added the tunning processes in the Supplement as Text S2:

"There are two categories of hyperparameters to be tuned, including hyperparameters for model structure (i.e., hidden layer, hidden size) and hyperparameters for the training

process (i.e., learning rate, batch size) (Li et al., 2024). Given a balance of model performance and time cost, 50 basins are selected randomly for the tuning processes. For each catchment, the first 14 years (from 1 October 1980 to 30 September 1994) of the entire training period (from 1 October 1980 to 30 September 1995) is set as the training period for tuning process and the last year (from 1 October 1994 to 30 September 1995) is set as the validation period. Models are trained using the Adam optimizer and the early stopping strategy. Three repetitions of each hyperparameter setting are used with different random seeds for initializing the weights, the mean NSE on validation period over the 3 repetitions represents the validation performance of a basin. Hyperparameters were chosen using the model settings with the highest median NSE scores on validation period over the 50 basins.

Firstly, the model structure hyperparameters are fine-tuned. Based on the model structure of Kratzert et al. (2018) (two hidden layers with the hidden size of 20) and the results that a one-layer LSTM network is qualified to capture rainfall-runoff response of a catchment (Kratzert et al., 2019, 2021), the range of the hidden size in this paper is set to 20,40,50,60, 80 and 100 while the number of hidden layers is set to 1. Following other hyperparameters of Kratzert et al. (2018), the LSTM is developed with input sequence for the past $T = 365$ d, a mini-batch size of 512, a drop-out rate of 0.1 and the Adam optimizer with a learning rate of 0.0001. Secondly, the hyperparameters for the training process is optimized based on the optimal hyperparameters in the first step. The LSTM network is tuned with different batch sizes (128, 256, 512), different learning rates (0.1, 0.01, 0.001, 0.0001), different learning rate decay (0.1, 0.3, 0.5, 0.7) and different dropout rates (0.2, 0.4, 0.6, 0.8).

After tunning, the optimal hyperparameters of the LSTM are shown by Table 1. In order to compromise between maximum reducing the uncertainty caused by different numbers of model parameters and achieving potentially more powerful predictions, the hidden sizes of the MC-LSTM network is set to 50, respectively, so that the numbers of parameters for MC-LSTM and LSTM differ by less than 0.1%. As the EXP-HYDRO model is a process-based model, there is no need for the DL wrapped EXP-HYDRO

model to normalize their input variables and to set the hidden size or dropout rate. Excluding the hidden size and dropout rate, the MC-LSTM and EXP-HYDRO models in the four experiments have the same hyperparameters as the LSTM, as shown by Table 1. Notably, the MC-LSTM has some hyperparameters from the LSTM instead of being optimized, while tunning the hyperparameters of the MC-LSTM can obtain better MC-LSTM networks. However, this paper aims to investigate the robustness of the LSTM and MC-LSTM rather than thoroughly explore the potential of the water balance constraint on the use of the LSTM network for rainfall-runoff prediction. Thus, further tuning processes of the MC-LSTM are not performed. Furthermore, the sensitivity analysis of model hyperparameters are devised based on model hyperparameters in Frame et al. (2023, 2022), so the hidden sizes of the LSTM and MC-LSTM are 256 and 64, respectively. The results of the sensitivity analysis of model hyperparameters are presented by Fig. S3 to S8 in the Supplement."

In addition, to consider the results of the experiments in this paper might be influenced by model hyperparameters, we have devised an additional experiment to demonstrate the reliability of the results in the Discussion section:

"Besides, the results of sensitivity analysis of model hyperparameters are presented by Fig. S3 to S8 in the Supplement. It can be observed that the enhanced robustness of the MC-LSTM compared with the LSTM changes little with different model hyperparameters, which demonstrates the reliability of the results in this paper." (Page 19, Lines 374 to 377)

[Figure]

Figure S3. As for Fig. 2, but for the LSTM and MC-LSTM with hidden sizes of 256 and 64, respectively, in 50 randomly selected basins.

[Figure]

Figure S4. As for Fig. 4, but for the LSTM and MC-LSTM with hidden sizes of 256 and 64, respectively, in 50 randomly selected basins.

[Figure]

Figure S5. As for Fig. 5, but for the LSTM and MC-LSTM with hidden sizes of 256 and 64, respectively, in 50 randomly selected basins.

[Figure]

Figure S6. As for Fig. 6, but for the LSTM and MC-LSTM with hidden sizes of 256 and 64, respectively, in 50 randomly selected basins. The MC-LSTM tends to be more stable at a total of 50 (100%), 43 (86%) and 45 (90%) basins when models are trained with data of 3 years, 9 years and 15 years, respectively.

[Figure]

Figure S7. As for Fig. 7, but for the LSTM and MC-LSTM with hidden sizes of 256 and 64, respectively, in 50 randomly selected basins.

[Figure]

Figure S8. As for Fig. 8, but for the LSTM and MC-LSTM with hidden sizes of 256 and 64, in 50 randomly selected basins. In the D/W scenario, the MC-LSTM exhibits higher KGE values compared to the LSTM across 38 basins (76%). But for the W/D scenario, the number of basins with higher KGE for the MC-LSTM than the LSTM decreases to 26 (52%).

**References:**

Frame, J. M., Kratzert, F., Klotz, D., Gauch, M., Shalev, G., Gilon, O., Qualls, L. M., Gupta, H. V., and Nearing, G. S.: Deep learning rainfall–runoff predictions of extreme events, Hydrol. Earth Syst. Sci., 26, 3377–3392, https://doi.org/10.5194/hess-26-3377-2022, 2022.

Frame, J. M., Kratzert, F., Gupta, H. V., Ullrich, P., and Nearing, G. S.: On strictly enforced mass conservation constraints for modelling the rainfall-runoff process, Hydrol. Process., 37, e14847, https://doi.org/10.1002/hyp.14847, 2023.

Kratzert, F., Klotz, D., Brenner, C., Schulz, K., and Herrnegger, M.: Rainfall-runoff modelling using long short-term memory (LSTM) networks, Hydrol. Earth Syst. Sci., 22, 6005–6022, https://doi.org/10.5194/hess-22-6005-2018, 2018.

Kratzert, F., Klotz, D., Herrnegger, M., Sampson, A. K., Hochreiter, S., and Nearing,

G. S.: Toward improved predictions in ungauged basins: exploiting the power of machine learning, Water Resour. Res., 55, 11344–11354, https://doi.org/10.1029/2019WR026065, 2019.

Kratzert, F., Klotz, D., Hochreiter, S., and Nearing, G. S.: A note on leveraging synergy in multiple meteorological data sets with deep learning for rainfall-runoff modeling, Hydrol. Earth Syst. Sci., 25, 2685–2703, https://doi.org/10.5194/hess-25-2685-2021, 2021.

Li, L., Dai, Y., Wei, Z., Shangguan, W., Zhang, Y., Wei, N., and Li, Q.: Enforcing Water Balance in Multitask Deep Learning Models for Hydrological Forecasting, J. Hydrometeorol., 25, 89–103, https://doi.org/10.1175/JHM-D-23-0073.1, 2024.

Nearing, G., Kratzert, F., Klotz, D., Hoedt, P.-J., Klambauer, G., Hochreiter, S., and Gupta, H.: A deep learning architecture for conservative dynamical systems: application to rainfall-runoff modeling, in: AI for Earth Sciences Workshop, NeurIPS 2020, 2020.

Wi, S. and Steinschneider, S.: On the need for physical constraints in deep learning rainfall–runoff projections under climate change: a sensitivity analysis to warming and shifts in potential evapotranspiration, Hydrol. Earth Syst. Sci., 28, 479–503, https://doi.org/10.5194/hess-28-479-2024, 2024.

---

## Author Comment (AC2)

**Response**

*Anonymous Referee #2:*

*General Comments*

*The authors derive a clear objective, investigating the robustness of LSTM and MC_LSTM against data sparsity, stability against parameter initialization, and test the transferability under different climatic conditions. The paper is in my opinion highly relevant given the current developments in using KI in Hydrology, it is generally well structured, easy to read and understandable and compact without missing relevant information. I belief therefore the manuscript is well suited for publication in the HESS journal.*

Thank you very much. We appreciate the positive comments.

*Some comments/suggestion that I believe would improve the manuscript and that should be addressed before final publication is the following:*

Thank you very much for the constructive suggestions and we have improved the paper accordingly. Below please find the point-to-point responses.

- *It is clearly stated and shown in the last publications of the Kratzert/Nearing group that the full potential of LSTM application can be achieved when training the LSTM on a large number of variable catchments including also static and dynamic catchment features (see also most recent contribution: https://eartharxiv.org/repository/view/6363/). I would at least like to see a discussion of this topic and how this is related to the presented work.*

Thank you very much for the insightful comment. Following the recommend literature, we have rewritten the Discussion section to clarify the differences and connections between this paper with the mentioned literature above. The related part are as follows:

"Although TGDS models can provide more accurate and robust predictions than pure DL models in basin-wise scale or data scarce conditions, it deserves additional scrutiny when trained with data from a large number of diverse basins (Frame et al., 2022; Nearing et al., 2021; Wi and Steinschneider, 2022). Recent studies have illustrated that the LSTM network works better for rainfall-runoff prediction when trained with a large amount of hydrologically diverse data than with data from a single watershed (Kratzert et al., 2024). Specifically, for DL models, physical constraints are effective in local models but offer little improvement in the regional models (Frame et al., 2023; Xie et al., 2021), even reduce predictive performance under extreme events (Frame et al., 2022). This outcome can be attributed to that pure DL models might be flexible enough to capture the behaviour in observation data with inconsistent water balance closure better than DL models constrained by the strict water balance (Kratzert et al., 2024; Frame et al., 2023; Beven, 2020). Besides, catchments with similar flood generating processes and similar characteristics may have some similar outliers and DL models can capture the rainfall-runoff responses among these basins (Xie et al., 2021; Bertola et al., 2023; Wi and Steinschneider, 2024). Therefore, there seems to be a compensating effect between data and knowledge on DL models, where the process knowledge is crucial for models trained with sparse data but less important with sufficient data. Large-sample hydrology is thus expected to enhance the performances of DL models for extreme events predictions and climate change projections (Bertola et al., 2023; Wi and Steinschneider, 2022, 2024).

Given that data is not always sufficient, the sensitivity of DL models when given scarce training data is essentially important (Feng et al., 2021; Gauch et al., 2021b). The TGDS provides effective tools for reducing data requirements of DL models (Karniadakis et al., 2021; Karpatne et al., 2017; Read et al., 2019; Xie et al., 2021). Therefore, this paper explores the effects of the water balance constraint on the robustness of the LSTM under restricted conditions, thereby training single model for each single basin rather than simultaneously for a large number of basins. Although the latter can achieve better performance (Kratzert et al., 2024), it is beyond the scope of

this paper but worthy of further study.…" (Page 19, Lines 378 to 397)

- *Given for example the results of Figure 2, they can be interpreted as LSTM's being "better" than the EXP-Hydro model. However, as actually mentioned by Beven (2020, doi10.1002/hyp.13805), it is still ~50% of the catchments show KGE-values of below 0.6, in my opinion indicating strong problems in the modelling outside the model-structure and calibration procedure.*

Thank you very much for the constructive comment. The catchments with KGE values below 0.6 are mainly located in the central arid regions of the United States. In these areas, the runoff generation is dominated by the infiltration-excess overland flow, which is largely controlled by short-duration, high-intensity precipitation events (Berghuijs et al., 2016). The EXP-HYDRO model operates with a mechanism of saturation-excess overland flow and is mainly applicable in the central arid regions (Jiang et al., 2020). Due to the infrequent storms and flood records in such regions, it is difficult for the LSTM network to satisfactorily reproduce flashy hydrographs (Jiang et al., 2020). Such a spatial pattern of model performance was also revealed in previous studies (Kratzert et al., 2018; Newman et al., 2017; Jiang et al., 2020). To dig into the applicable model in such regions is beyond the scope of this paper but worthy of further study. In order to show the competitive performances of the LSTM, MC-LSTM and EXP-HYDRO models in this paper compared to previous studies, we have added a comparison table about the model performance:

"Table 2 compares the Nash-Sutcliffe efficiency (NSE) of the three models in this paper with those in previous studies (Jiang et al., 2020; Kratzert et al., 2018; Patil and Stieglitz, 2014; Newman et al., 2017; Hoedt et al., 2021), most of which are based on the same dataset and the roughly overlapping testing period as those in this paper. Among these models, the LSTM, MC-LSTM and EXP-HYDRO models in this paper exhibit competitive performances, suggesting the reasonability of the hyperparameter optimization and calibration procedures." (Page 11, Lines 247 to 252)

**Table 2.** Comparison of daily NSE statistics across the CAMELS catchments.

| Model | Single model for | Count of basins | Dataset | Daily NSE statistics | | | Source |
| --- | --- | --- | --- | --- | --- | --- | --- |
| | | | | median | mean | Proportion for NSE ≥0.55 | |
| LSTM | Single basin | 531 | CAMELS | 0.67 | 0.63 | 76% | This paper |
| MC-LSTM | Single basin | 531 | CAMELS | 0.63 | 0.59 | 71% | This paper |
| EXP-HYDRO* | Single basin | 531 | CAMELS | 0.49 | 0.42 | 40% | This paper |
| LSTM | Single basin | 569 | CAMELS | 0.60 | 0.52 | 61.5% | Jiang et al. (2020) |
| EXP-HYDRO* | Single basin | 569 | CAMELS | 0.48 | -0.16 | 38.3% | Jiang et al. (2020) |
| LSTM | Single basin | 241 | CAMELS | 0.65 | 0.63 | NA | Kratzert et al. (2018) |
| EXP-HYDRO | Single basin | 756 | HCDN | NA | NA | ~43% (>0.6) | Patil and Stieglitz (2014) |
| VIC | Single basin | 531 | CAMELS | 0.57-0.59 | NA | ~56% | Newman et al. (2017) |
| LSTM | Multiple basins | 447 | CAMELS | 0.737 | NA | NA | Hoedt et al. (2021) |
| MC-LSTM | Multiple basins | 447 | CAMELS | 0.726 | NA | NA | Hoedt et al. (2021) |

*HCDN: Hydro-Climate Data Network; VIC: Variable Infiltration Capacity model*

*EXP-HYDRO*: Deep learning wrapped EXP-HYDRO model; NA: not available*

Additionally, as mentioned by the literature (Beven, 2020), the observation data with inconsistent water balance closure may be one of the reasons for that physical constraints offer little improvement in the regional models and even reduce predictive performance under extreme events, thus we have added this reason in the Discussion section:

"This outcome can be attributed to that pure DL models might be flexible enough to capture the behaviour in observation data with inconsistent water balance closure better than DL models constrained by the strict water balance (Kratzert et al., 2024; Frame et al., 2023; Beven, 2020)" (Page 19, Lines 384 to 386)

- *I believe the statement in L374-375 is not supported by the experimental design of the paper – no LSTM model is trained simultaneously to many catchments here, so the statement needs to be modified – or I have misread section 2/3*

Thank you very much for the careful comment. We are sorry for the confusing information and have removed this sentence.

*Specific/technical Comments*

*The following minor comments/suggestions I would like to make:*

- *L51: "On" instead of "One"*

Thanks a lot for your careful comment. We have corrected this mistake.

- *L53: Mass Balance has already been introduced by Frame et al. 2023*

Thank you for the insightful comment. We have added this citation:

"On the one hand, without explicit physical mechanism such as the conservation of mass and energy, the LSTM network cannot guarantee causal relationships as physical models can (Wang et al., 2023; Xie et al., 2021; Frame et al., 2023), which may lead to spurious and inaccurate prediction that is potential to violate water balance, particularly when extrapolating beyond the range of training data (Bhasme et al., 2022; Reichstein et al., 2019)." (Page 2, Lines 45 to 49)

- *L66: please define robustness as used here – in statistics it has a very specific meaning related to performance when a priori assumption (e.g. Normality) are violated*

Thank you for the constructive comment. The definition of the robustness used in this paper has been added:

"Therefore, there is yet no consensus on the effects of the water balance constraint on the use of the LSTM network for rainfall-runoff prediction (Pokharel et al., 2023), particularly its robustness—the ability to perform consistently across varying conditions (Manure et al., 2023). Aiming to bridge the gap, this paper focuses on how the water balance constraint in model architecture affects the robustness of the basin-wise trained LSTM network for rainfall-runoff prediction. Focusing on the robustness

of the LSTM and MC-LSTM, the objectives are to examine (1) the sensitivity to data sparsity, (2) the stability against random parameters initialization and (3) the transferability under contrasting climate conditions…" (Page 3, Lines 60 to 66)

- *L80: a small figure as e.g. in Kratzert et al. 2018 to visualize the LSMT would not be bad, the equations are not intuitive, it would also help in L98f to understand the implementation of the MC*

Thanks for your suggestion. To help understand the implementation of mass balance, two figures of the internal operations of a standard LSTM network and a MC-LSTM network have been added in the Supplement as Fig. S1 and Fig. S2, respectively.

[Figure]

Figure S1. The internal operation of a standard LSTM network.

[Figure]

Figure S2. The internal operation of a MC-LSTM network.

- *L120: Equation 11 does not explain how M, ET Q Ps and Pr are calculated – can also go into an appendix*

Thank you. The explanation of how to calculate the 5 flux variables of the EXP-HYDRO model has been added in the Supplement as Text S1:

"The EXP-HYDRO model is a conceptual, spatially lumped rainfall-runoff model developed by Patil and Stieglitz (2014). The physical equations and parameters are well introduced and organized in Text S1 in the Supporting Information of Jiang et al. (2020). For easy reading, the calculation equations of the 5 flux variables ($M$, $ET$, $Q$, $P_s$ and $P_r$) are briefly introduced here.

$P_s$ and $P_r$ are respectively the daily snowfall (mm/day) and rainfall (mm/day), which are estimated by the daily precipitation ($P$, mm/day) and daily temperature ($T$, °C) as follows:

$$P_s = fun1(P, T, T_{min}) = \begin{cases} 0, & T > T_{min} \\ P, & T \le T_{min} \end{cases} \tag{S1}$$

$$P_r = fun2(P, T, T_{min}) = \begin{cases} P, & T > T_{min} \\ 0, & T \le T_{min} \end{cases} \tag{S2}$$

Where $T_{min}$ is a parameter representing the temperature threshold where the precipitation falls as snow.

The snowmelt ($M$, mm/day) is simulated by a simple thermal degree-day model related to $T$ and the snow accumulation bucket ($S_0$) based on the following equation:

$$M = fun3(S_0, T, D_f, T_{max}) = \begin{cases} min\{S_0, D_f \cdot (T - T_{max})\}, & T > T_{max} \text{ and } S_0 > 0 \\ 0 & otherwis \end{cases}$$

Where $D_f$ is a parameter denoting the thermal degree-day factor (mm/day/°C); $T_{max}$ is another parameter representing the temperature threshold where the accumulated snow begins to melt.

The evapotranspiration is denoted by $ET$ (mm/day), which is calculated as the fraction of the potential evapotranspiration ($PET$, mm/day, estimated by Hamon's formulation as Eq. S5) as follows:

$$ET = fun4(S_1, PET, S_{max}) = \begin{cases} 0, & S_1 < 0 \\ PET \cdot \left(\dfrac{S_1}{S_{max}}\right), & 0 \le S_1 \le S_{max} \\ PET, & S_1 > S_{max} \end{cases} \tag{S4}$$

$$PET = 29.8 \cdot L_{day} \cdot \frac{e^*(t)}{T + 273.2} \tag{S5}$$

$$e^*(t) = 0.611 \cdot e^{17.3 \cdot T/(T+237.3)} \qquad \text{(S6)}$$

Where catchment bucket $(S_1)$ denotes its current storage; $S_{max}$ is a parameter representing the storage capacity of the catchment bucket; $L_{day}$ is the day length (hour).

The streamflow $(Q)$ is estimated as the sum of the baseflow $(Q_b)$ and the capacity-excess runoff $(Q_s)$, which are respectively expressed as follows:

$$Q_b = fun5(S_1, f, S_{max}, Q_{max}) = \begin{cases} 0, & S_1 < 0 \\ Q_{max} \cdot e^{-f \cdot (S_{max} - S_1)}, & 0 \leq S_1 \leq S_{max} \\ Q_{max}, & S_1 > S_{max} \end{cases} \qquad \text{(S7)}$$

$$Q_s = fun6(S_1, S_{max}) = \begin{cases} 0, & S_1 \leq S_{max} \\ S_1 - S_{max}, & S_1 > S_{max} \end{cases} \qquad \text{(S8)}$$

$$Q = Q_b + Q_s \qquad \text{(S9)}$$

Where $f$ and $Q_{max}$ are two parameters representing the decline rate of runoff (mm$^{-1}$) and the maximum subsurface runoff (mm/day), respectively."

- *L125: 1-2 sentences on how EXP-Hydro is wrapped into a DL architecture would be interesting*

Thank you very much for the constructive suggestion. We have added more details of the DL-wrapped EXP-HYDRO model and revised the original information for easy understanding:

"That is, the EXP-HYDRO model is rewritten using a differentiable PyTorch framework (Paszke et al., 2019), the typical recurrent neural network architecture, where the mathematical expressions and learnable parameters is replaced with the physical equations and parameters of the EXP-HYDRO model (Zhong et al., 2023; Jiang et al., 2020)." (Page 5, Lines 120 to 124)

- *L209: I do not think "maximumly" can be used – using to a maximum!?*

Thanks a lot for your careful comment. We have corrected this mistake.

- *L423: there are no further co-authors!*

Thanks a lot for your careful comment. We have corrected this mistake.

I feel, the manuscript has in general the potential to be a valuable contribution to HESS, however, questions and issues raised in the general comments would need to be addressed and discussed to a significant part before final acceptance.

We greatly appreciate your positive comments and constructive suggestions.

**References**

Berghuijs, W. R., Woods, R. A., Hutton, C. J., and Sivapalan, M.: Dominant flood generating mechanisms across the United States, Geophys. Res. Lett., 43, 4382–4390, https://doi.org/10.1002/2016GL068070, 2016.

Hoedt, P.-J., Kratzert, F., Klotz, D., Halmich, C., Holzleitner, M., Nearing, G. S., Hochreiter, S., and Klambauer, G.: MC-LSTM: mass-conserving LSTM, in: Proceedings of the 38th International Conference on Machine Learning, International Conference on Machine Learning, 4275–4286, 2021.

Jiang, S., Zheng, Y., and Solomatine, D.: Improving AI system awareness of geoscience knowledge: symbiotic integration of physical approaches and deep learning, Geophys. Res. Lett., 47, e2020GL088229, https://doi.org/10.1029/2020GL088229, 2020.

Kratzert, F., Klotz, D., Brenner, C., Schulz, K., and Herrnegger, M.: Rainfall-runoff modelling using long short-term memory (LSTM) networks, Hydrol. Earth Syst. Sci., 22, 6005–6022, https://doi.org/10.5194/hess-22-6005-2018, 2018.

Newman, A. J., Mizukami, N., Clark, M. P., Wood, A. W., Nijssen, B., and Nearing, G.: Benchmarking of a Physically Based Hydrologic Model, Journal of Hydrometeorology, 18, 2215–2225, https://doi.org/10.1175/JHM-D-16-0284.1, 2017.

Patil, S. and Stieglitz, M.: Modelling daily streamflow at ungauged catchments: what

information is necessary?, Hydrol. Process., 28, 1159–1169, https://doi.org/10.1002/hyp.9660, 2014.